# Evaluation of Ultrasound-Based Point Shear Wave Elastography for Differential Diagnosis of Pancreatic Diseases

**DOI:** 10.3390/diagnostics12040841

**Published:** 2022-03-29

**Authors:** Bozhidar Hristov, Vladimir Andonov, Daniel Doykov, Silvia Tsvetkova, Katya Doykova, Mladen Doykov

**Affiliations:** 1Second Department of Internal Diseases, Section “Gastroenterology”, Medical Faculty, Medical University of Plovdiv, 6000 Plovdiv, Bulgaria; vladkoandonov@abv.bg (V.A.); daniel_doykov@abv.bg (D.D.); 2Gastroenterology Clinic, University Hospital “Kaspela”, 4001 Plovdiv, Bulgaria; 3Department of Diagnostic Imaging, Medical Faculty, Medical University of Plovdiv, 6000 Plovdiv, Bulgaria; sts2001@abv.bg (S.T.); katya.doykova@mu-plovdiv.bg (K.D.); 4Department of Diagnostic Imaging, University Hospital “Kaspela”, 4001 Plovdiv, Bulgaria; 5Department of Urology and General Medicine, Medical Faculty, Medical University of Plovdiv, 6000 Plovdiv, Bulgaria; mladen.doykov@mu-plovdiv.bg; 6Clinic of Urology, University Hospital “Kaspela”, 4001 Plovdiv, Bulgaria

**Keywords:** pancreatitis, pancreatic carcinoma, elastography, pSWE

## Abstract

Introduction: A variety of imaging techniques exists for the diagnosis of pancreatic disorders. None of the broadly applied diagnostic methods utilizes elasticity as an indicator of tissue damage. A well-known fact is that both chronic pancreatitis (CP) and pancreatic ductal adenocarcinoma (PDA) are associated with the development of prominent fibrosis (increased tissue stiffness). Purpose: To prospectively assess the accuracy of point shear wave elastography (pSWE) in differentiating between benign and malignant pancreatic diseases, establish a cut-off value for the diagnosis of PDA, and evaluate the influence of certain variables on the obtained results. Materials and methods: The present study included 78 patients who were admitted at the Department of Gastroenterology at the university hospital “Kaspela” between December 2017 and August 2021 for diagnosis and/or treatment of pancreatic disorders. Based on the clinical criteria, diagnostic imaging, and histological findings, patients were divided into the CP and PDA group. The ultrasound based pSWE technique was applied and shear wave velocity (SWV) was measured. The depth of region of interest (ROI) and successful measurement rate were also recorded. Results: The mean ± SD SWV values established through pSWE were 1.75 ± 0.34 m/s and 2.93 ± 0.91 m/s for the CP and PDA, respectively. With a cut-off value of 2.09 m/s, we calculated the sensitivity (Se), specificity (Sp), and accuracy for differentiating between CP and PDA of 89.47%, 91.20%, and 88.60%, respectively. Of the examined variables, BMI and depth of ROI in the CP group and sex in the PDA group showed a statistically significant influence on the obtained results. Conclusions: pSWE may be utilized as a differential diagnostic modality in patients with suspected CP or PDA.

## 1. Introduction

Pancreatic disorders represent a considerable health issue for contemporary medicine. In 2020, 132,134 deaths due to pancreatic cancer were registered in Europe. The incidence of the disease is expected to rise by 40% until 2035 [1]. By 2030, PDA is expected to be the second most common cause of cancer related death. Quite worrisome is the fact that instead of declining, the overall mortality has increased by 53% in the last 25 years [2]. These facts are largely due to the relatively late diagnosis of the disease. Research shows that in only 9% of patients is the tumor confined to the pancreas at the time of the diagnosis and in all others, locally advanced or metastatic disease is established. The five-year survival rate in the latter does not exceed 2% [3]. The health burden of chronic pancreatitis also cannot be overestimated. Epidemiological studies show an annual incidence of CP of about 7.8/100,000 people. This fact, and the average life expectancy of patients with CP of about 18–20 years, have set a global prevalence of CP at about 120–143/100,000 [4]. Even discounting the short-term potential life threatening events, in the long run, CP is accompanied by debilitating events including pain, malnutrition, development of diabetes, and pancreatic cancer. The obstacles in the successful management of CP are largely due to the almost universally late diagnosis of the disease, which in turn is determined by the lack of reliable criteria for histological evaluation and the widely available imaging techniques for early diagnosis. Table 1 presents the currently available imaging techniques for the diagnosis of pancreatic disorders with their respected sensitivity and specificity (Table 1) [5,6,7].

Though generally reliable, existing imaging modalities have proved to be insufficient in a certain context. CT scan is associated with not negligible radiation exposure. Its sensitivity for the early forms of CP is unsatisfactory. Moreover, in about 11% of the patients, differentiation between CP and PDA is impossible based on CT [8]. MRCP has high Se and Sp for the diagnosis of pancreatic disorders. Unfortunately, the lack of availability and need for highly qualified medical personnel narrow its broad usage in real clinical practice. EUS is considered to be the most reliable method for early diagnosis of CP. Its invasiveness, and furthermore its subjective nature, leads to unacceptably low interobserver agreement (IOA), compromising its reproducibility. ERCP was considered in the past as the “gold standard” for diagnosis of pancreatic diseases, but has nowadays been largely abandoned as a diagnostic method due to its considerable complication rate.

It is a scientifically proven fact that both CP and PDA are associated with prolific fibrogenesis. Pancreatic fibrogenesis in CP closely resembles the pathologic process seen in liver parenchyma. The chief histological feature of PDA is the presence of prominent desmoplastic reaction characterized by the excessive proliferation of fibroblasts and the accumulation of extracellular matrix. In both cases, this results in increased stiffness (decreased elasticity) of the pathological regions compared to the normal pancreatic parenchyma.

Taking this fact into consideration, the introduction of ultrasound elastography is naturally considered to be an encouraging new step in the diagnosis of pancreatic disorders. US elastography is an ultrasound-based method that evaluates tissues in terms of their elasticity, which is an entirely new approach to the characterization and diagnosis of various diseases. At the time of its creation, US elastography was largely inapplicable to the pancreas and it was impossible to evaluate the elasticity of an organ so distant to the body surface. The introduction of shear wave elastographic techniques, particularly of pSWE, radically changed this postulate. pSWE, as part of the so called dynamic elastographic modalities, measures tissue stiffness on the basis of the so called “shear waves”. A pSWE ultrasound beam emitted by standard convex transducer generates displacement in a certain region of interest (region of excitation), which usually measures 10 × 5 mm. Displacement on its behalf generates waves in the parenchyma perpendicular to the ultrasound beam called “shear waves”. The speed of the SW is established using so called “tracing beams” emitted by the transducer that measure the spread of the waves between two points over a certain period of time. In real life, the correlation between tissue stiffness and shear wave velocity is linear and tissue elasticity is practically the only determinant of SWV. Research evaluating the usage of pSWE for the diagnosis of pancreatic disorders is scarce.

The aim of the present study was to prospectively assess the accuracy of pSWE for differentiation between benign (CP) and malignant (PDA) pancreatic diseases. As a secondary endpoint, we evaluated the influence of certain variables on the obtained results.

## 2. Materials and Methods

In the present study, we included 78 patients, admitted at the Department of Gastroenterology at the university hospital “Kaspela” for the diagnosis and/or treatment of pancreatic disorders from December 2017 to August 2021. Patients were divided into the CP and PDA group depending on the below listed criteria.

### 2.1. Group 1. Chronic Pancreatitis

#### 2.1.1. Inclusion Criteria

In this group, we reviewed 35 patients defined as having CP based on contrast enhanced CT (CECT). Inclusion criteria were based on the Cambridge criteria for CP adapted for CECT. Only patients with CECT findings consistent with CP were included in the study (Cambridge class II (mild) to IV (severe)). Patients with suspected CP (Cambridge class I) were considered ineligible for evaluation. Adequate visualization of the pancreatic gland by means of conventional B-mode was mandatory. Included patients were to have no signs of exacerbation of CP at least three months prior to the investigation, but overall had over three episodes of exacerbation. Serum amylase and lipase levels below three times upper normal range (UNR) were required. An extensive follow up over three months after the pSWE was conducted was to further mitigate the possibility of PDA misdiagnosed as CP (Table 2).

#### 2.1.2. Exclusion Criteria

Inadequate visualization of all parts of the pancreas by means of conventional B-mode. Current evidence of acute pancreatitis or exacerbation of chronic pancreatitis. Episode of pancreatitis less than three months prior to testing.

### 2.2. Group 2. Pancreatic Ductal Adenocarcinoma

#### 2.2.1. Inclusion Criteria

In this group, we included 38 patients with histologically proven pancreatic ductal adenocarcinoma. Histological specimens were obtained by means of percutaneous ultrasound guided biopsy (utilizing Tru-cut 18 Ga, 22 mm biopsy needle TSK Laboratory, Tochigi-Shi, Japan) or EUS-guided fine needle biopsy (FNB) (utilizing 22 Ga FNB needle, Acquire Boston Scientific, Marlborough, MA, USA). Percutaneous biopsy was performed in patients considered to be poor surgical candidates (locally advanced or metastatic disease), while EUS-FNB was used in potentially resectable tumors in order to minimize the risk of tumor seeding. Adequate visualization of the ROI was also accounted for when choosing the diagnostic modality.

#### 2.2.2. Exclusion Criteria

Inadequate visualization of the tumor or the remaining non-cancerous parenchyma by means of the conventional B-mode as well as the inability to obtain histological confirmation.

### 2.3. Examination Technique

All patients fasted for at least six hours and lay in the supine position at least ten minutes prior to the start of the examination. First, a conventional ultrasound of the abdomen was performed. Adequate visualization of the entire pancreas was a prerequisite for inclusion in the study. The size and structure of the pancreas, the presence of focal lesions, calcifications, and pancreatic fluid collections were recorded. The ductal system was also evaluated. Additionally, we looked for signs of metastatic disease, obstructive jaundice, liver cirrhosis, and associated portal hypertension and splenomegaly. The patient was in the supine position throughout the entire test. The convex transducer was positioned at the epigastrium and targeted toward the celiac trunk. The splenic vein was used as the main landmark and presented by positioning the transducer in the caudal direction. Successively, the head and the tail of the pancreas were evaluated by placing the transducer respectively downward and to the right and upward and to the left. After completion of standard B-mode, pSWE was conducted and SWV was measured. Five valid measurements were required in every part of the pancreas in patients with CP. In patients with PDA, ten valid measurements were performed at the site of the lesion and ten in the surrounding tumor free parenchyma. The depth of ROI was also recorded. To conduct the examination, we used a Siemens Acuson S2000 ultrasound machine equipped with a 6C1 HD transducer 1.5–6 MHz using virtual touch quantification (VTQ) software. Invalid results or such exceeding the range of the device were presented as X.XX m/s and therefore excluded from the analysis. 

### 2.4. Statistical Methods

Obtained results were collected in a Microsoft Excel table. Statistical analysis was conducted using IBM Statistical Package for Social Sciences (SPSS) (24th Version) and MedCalc, version 20.014, 2021. Metric values symmetry was checked by using the Kolmogorov–Smirnov test. In the presence of normal distribution (*p* > 0.05), metric variables were presented with the mean value and standard deviation and were analyzed by means of parametrical statistical methods. In the absence of normal distribution (*p* < 0.05), we utilized median values, interquartile range (IQR), and non-parametrical statistical methods. Correlation between qualitive variables was studied by means of Chi-square analysis. To measure the accuracy of pSWE for differentiation between benign and malignant pancreatic diseases, we investigated the area under the receiver operating characteristic curve (AUROC). Adequate cut-off values were selected to achieve optimal Se and Sp. A confidence interval (CI) of 95% was adopted.

## 3. Results

In the present study, we evaluated 78 patients: 39 with PDA and 39 with CP. In five patients (6.4%) (four with CP and one with PDA), adequate visualization of the entire pancreatic gland was impossible, so were therefore excluded from further analysis. In total, a sufficient number of valid measurements of SWV were obtained in 93.5% of the patients in the head, 97.2% in the body, and 92.8% in the tail of the pancreas.

### 3.1. General Characteristics of the Studied Groups

CP group: There was a statistically significant prevalence of male sex in the CP group (*p* = 0.004) 68.60% vs. 31.40% (24 males, 11 females). Median age was 61 years with age distribution being irrespective of sex (*p* = 0.109). Majority of the patients had normal BMI (57.10%), with 28.60% having a BMI above 25 kg/m^2^, but only 6.8% being obese (BMI above 30 kg/m^2^) and none being extremely obese (BMI above 35 kg/m^2^).

PDA group: Sex distribution was equal in the evaluated group (male:female = 19:19). Median age was 65.34 years with no statistical difference between sexes (*p* = 0.731) and with the CP group. Equal distribution of BMI and tobacco smoking was observed in males and females. Again, normal BMI was prevalent (71.10%), with low rates of obese and extremely obese patients (6.4%).

### 3.2. CP Group

Five valid measurements were performed at the head, body, and tail of the pancreas. Depth of ROI was also recorded. Mean ± SD, median, min.–max., and reference values were estimated for the head, body, and tail of the pancreas. Since there was no statistically significant difference between the obtained results, identical values for the entire pancreatic parenchyma were calculated. Results are presented in Table 3 and Table 4.

Mean ± SD value for the entire pancreatic parenchyma was established at 1.75 ± 0.34 m/s, with reference values of 0.99 m/s to 2.43 m/s. To assess the influence of certain variables on the obtained results, we investigated the correlation of SWV with factors such as depth of ROI, age, sex, BMI, smoking, alcohol consumption, diabetes, and stage of CP. Of all the evaluated factors, depth of ROI (*p* = 0.004) and BMI (*p* = 0.015) showed statistically significant negative association with SWV values. Results are presented in Figure 1 and Figure 2.

We established that severe CP (Cambridge class IV) was consistently associated with higher SWV values of 1.86 m/s vs. 1.58 m/s and 1.66 m/s for mild and moderate pancreatitis, respectively (*p* = 0.036).

### 3.3. PDA Group

Ten valid measurements were performed at the tumor and the surrounding healthy parenchyma, respectively. A total of 81.6% (31) of the tumors were located in the head, 15.8% (6) in the body, and 2.6% (1) in the tail of the pancreas. Depth of ROI was recorded. Mean ± SD, min.–max., and reference values were estimated for the region of the tumor and the surrounding non-cancerous parenchyma. Since a statistically significant difference (*p* = 0.009) between SWV in males and females was found, the respective results for both sexes were calculated. Results are presented in Table 5.

To assess the influence of certain variables on the obtained results, we investigated the correlation of SWV with factors such as depth of ROI, age (*p* = 0.085), sex, BMI (*p* = 0.480), smoking (*p* = 0.120), alcohol consumption (*p* = 0.114), diabetes (*p* = 0.191), and metastatic disease (*p* = 0.640). Of note, the association of SWV with the presence of obstructive jaundice (*p* = 0.546) was studied. None of the researched factors were proven to have a statistically significant influence on the SWV values. The correlation between SWV and obstructive jaundice is presented in Figure 3.

To establish the diagnostic capabilities of pSWE for differentiating between CP and PDA, we created a receiver operating characteristic curve (ROC). A confidence interval of 95% was adopted. A statistically significant difference was established between SWV values in CP and PDA (*p* < 0.001). Results are presented in Figure 4.

With a cut-off value of 2.09 m/s, we established an accuracy of pSWE for differentiating between CP and PDA of 88.60%, with Se of 89.47% and Sp of 91.43%. Results are presented in Table 6 and Figure 4.

## 4. Discussion

pSWE has been confirmed as a reliable diagnostic modality for evaluating the tissue stiffness of various abdominal structures [9] including the measurement of liver fibrosis [10]. However, few studies have investigated the applicability of the method on the pancreas. D’Onofrio at al. were the first to apply pSWE on the pancreas. Using pSWE, they successfully diagnosed a cystadenoma misdiagnosed as a solid neoplasm on CT and conventional ultrasonography [11]. Kawada et al. were the first to prove that pSWE was a viable option in most of the investigated patients [12]. In the present study, we confirmed this observation by performing successful measurements in all parts of the pancreas in more than 90% of the cases. Additionally, we found that BMI had little effect on the success rate of pSWE, since very few of the patients in the studied population were morbidly obese (6.4%). This is to be expected considering the fact that weight loss is a distinct feature of both CP and PDA.

In our study, we established mean ± SD and reference values of 1.75 ± 0.34 m/s and 0.99 m/s to 2.43 m/s, respectively, in patients with CP. Mateen et al. conducted a similar study establishing reference values for CP of 0.65 to 1.74 m/s and mean ± SD of 1.24 ± 0.234 m/s [13]. Those were considerably lower compared to the ones calculated in our research. A few facts, however, should be taken into consideration. First, patient selection in the aforementioned study was based on quite ambiguous criteria. For instance, any imaging modality suggestive of CP could be used as a diagnostic method including plain radiography in certain patients. Additionally, patients with local complications (advanced CP) were excluded from the analysis, naturally leading to lower SWV values. Llamoza-Torres et al. calculated the mean ± SD values for the entire pancreatic parenchyma of 1.57 ± 0.54 m/s and 1.67 ± 0.66 m/s, 1.57 ± 0.54 m/s, and 1.36 ± 0.45 m/s in the head, body, and tail, respectively [14]. These results were closer to those established in our study, but still quite a bit lower. This again could be explained by the patient selection. Llamoza-Torres et al. used EUS as the primary diagnostic modality and classified patients with CP according to the Rosemont classification. Of note, patients with suspected (not consistent with) CP were also included in the analysis, while ones with local complications were excluded.

In the present study, we only evaluated patients with findings consistent with CP (Cambridge II–III) and local complications were not regarded as exclusion criteria. This decision was based on the study design, which aimed as a secondary endpoint to evaluate the influence of stage of CP on the SWV values. Indeed, it was proven that SWV was strongly correlated with the CP stage. This observation is an important accomplishment since it justifies future research to prove that pSWE can be successfully used to diagnose even early forms of CP and to monitor the evolution of CP. Of note, though the presence of local complications (mainly chronic fluid collections) was not considered as exclusion criteria per se, we found that in the four patients with CP excluded from the analysis that universally, a large fluid collection precluding adequate visualization of the pancreas was present. This shows that even the results in our study were possibly underestimated.

Yashima et al. also found statistically significant elevation of SWV in CP compared to healthy subjects [15].

Regarding PDA, in our study, we established mean ± SD and reference values of 2.92 ± 0.91 m/s and 1.01–4.85 m/s for the SWV in the region of the tumor. These results are similar to the ones calculated in a study conducted by Park et al. of 3.3 ± 1.1 m/s [16]. Goertz at al. and Zaro et al. established that SWV was considerably higher in patients with PDA compared to healthy subjects [17,18].

In both groups, we assessed the influence of multiple variables on the measured SWV. To date, such studies are scarce and the obtained results are contradictory. In the PDA group, there was a statistically significant difference in SWV based on sex. In females, consistently higher SWV values were established. Similar research has not been conducted in any of the existing studies. The clinical relevance of such observations will need to be determined in future studies. Even if we account for this fact, the established SWV were universally and exceedingly higher in PDA patients compared to healthy subjects and even the CP group.

In CP, we found that depth of ROI and BMI were negatively associated with SWV values, meaning that lower BMI and more superficial location of the ROI corresponded to higher SWV. Kawada et al. [12] evaluated the depth of ROI and established the opposite results. Regarding BMI, it was assessed as a variable in research conducted by Pozzi et al. [19] and Yashima et al. [15], who obtained results consistent with ours. As already mentioned, the stage of CP showed significant correlation with the SWV values, however, this finding needs to be confirmed in future studies.

It is known that the chief histological feature of CP is progressive replacement of normal pancreatic parenchyma with fibrotic tissue. This process is accompanied by gradual reduction in exocrine function, which becomes clinically evident when more than 90% of the functional capacity of the pancreas is exhausted. Exocrine pancreatic insufficiency manifests with malnutrition and malabsorption, which, combined with the endocrine insufficiency will eventually result in considerable weight loss. That said, we speculate that the negative correlation of BMI and depth of ROI with SWV actually reflects the correlation between pancreatic fibrogenesis (main determinant of SWV) and loss of pancreatic function, causing decreased body weight (BMI) and loss of abdominal muscular and fat tissue, resulting in more superficial location of the gland.

Primary endpoint in our study was to evaluate the capability of pSWE to differentiate between CP and PDA. We found that with a cut-off value of 2.09 m/s, PDA could be successfully discerned from CP with accuracy, with a Se and Sp of 88.60%, 89.47%, and 91.43%, respectively. Currently, there are three studies investigating this subject. Goetz et al. established a cut-off value of 1.74 m/s for the diagnosis of PDA with Se of 91.1% and Sp of 60.4%. By increasing the cut-off value in our study, we immensely improved the specificity at the expense of very slight compromise in sensitivity. Park et al. [16] established mean ± SD SWV values of 2.4 ± 1.1 m/s and 3.3 ± 1.0 m/s for the CP and PDA group, respectively. Although the discrepancy in the results was obvious, it failed to achieve statistical relevance. Zaro et al. [18] are currently conducting a pilot study investigating pancreatic tumors by means of pSWE. Based on the preliminary results, they calculated a cut-off value of 1.54 m/s for the diagnosis of PDA, but data regarding accuracy, Se, and Sp are yet to be published.

## 5. Conclusions

pSWE could be successfully adopted as a diagnostic modality in patients with CP and PDA. It is superior to conventional US in B-mode for differentiating between benign and malignant pancreatic disorders and could be used as a readily available, cheap, and reproducible modality in this subgroup of patients. The main disadvantage of the method is the suboptimal approach to the pancreatic gland, which in this subgroup of patients is mostly associated with the presence of large fluid collections.

## Figures and Tables

**Figure 1 diagnostics-12-00841-f001:**
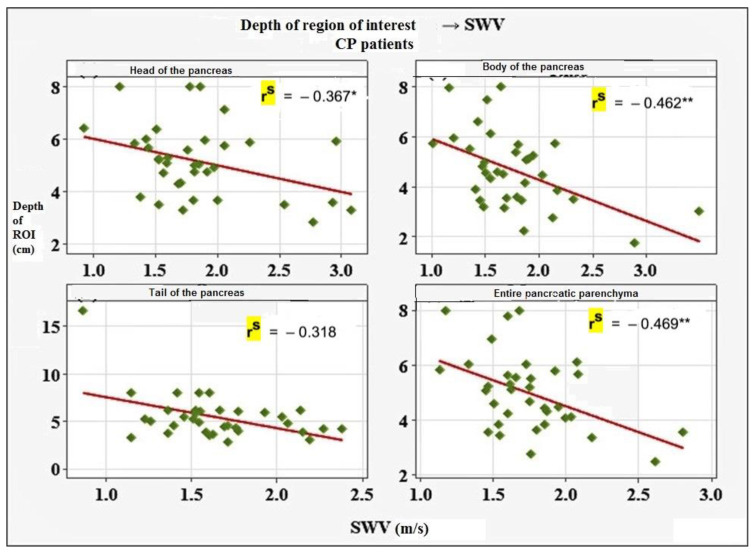
Association between depth of ROI and SWV in CP patients. *—significant association (*p* < 0.05); **—significant association (*p* < 0.01); rs—Spearman rank-order correlation (negative values re-flect inverse correlation).

**Figure 2 diagnostics-12-00841-f002:**
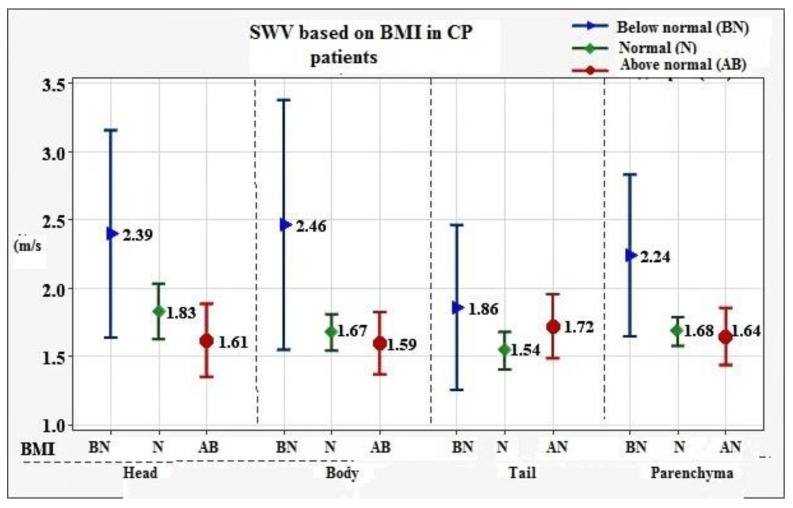
Association between BMI and SWV in CP patients.

**Figure 3 diagnostics-12-00841-f003:**
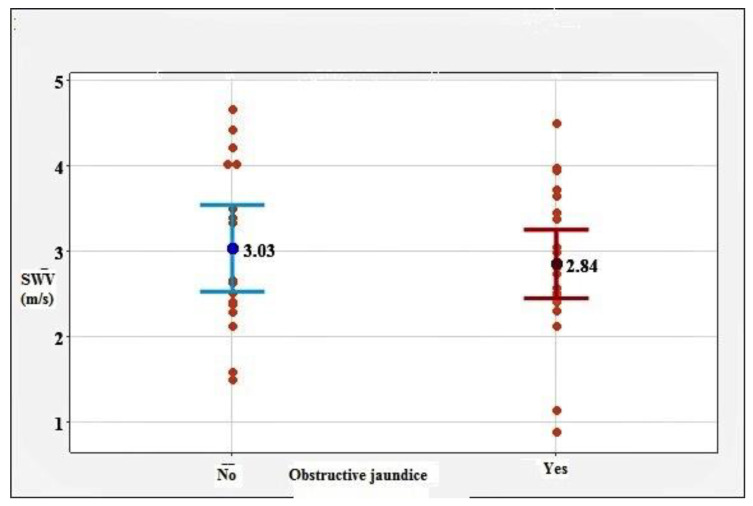
SWV values in correlation with obstructive jaundice.

**Figure 4 diagnostics-12-00841-f004:**
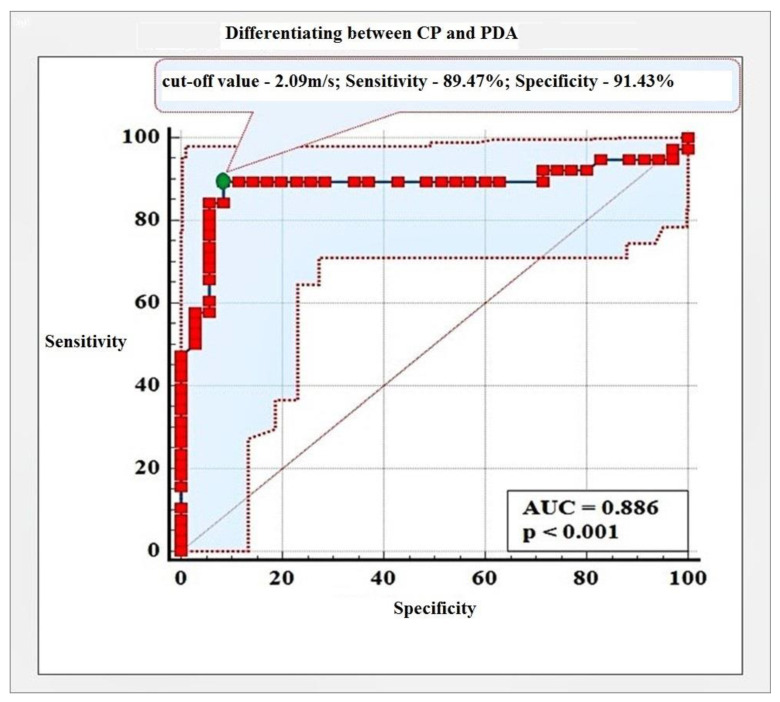
Performance of pSWE for differentiating between CP and PDA.

**Table 1 diagnostics-12-00841-t001:** Existing imaging techniques for the diagnosis of pancreatic disorders.

Imaging Modality	Chronic Pancreatitis		Pancreatic Cancer	
	Sensitivity (%)	Specificity (%)	Sensitivity (%)	Specificity (%)
Computer tomography (CT)	75%	91%	91–95%	100%
Endoscopic retrograde cholangiopancreatography (ERCP)	70–80%	80–10%	70%	94%
Magnetic resonance cholangiopancreatography (MRCP)	88%	98%	84–91%	97%
Ultrasonography (US)	60–81%	70–97%	76%	75%
Endoscopic ultrasonography (EUS)	80–100%	80–100%	98%	95.8%

**Table 2 diagnostics-12-00841-t002:** Cambridge classification of CP adapted for CT.

Cambridge 0 (Normal Pancreas)	Normal Pancreatic Parenchyma
Cambridge 1 (Uncertain)	It is impossible to exclude or confirm the diagnosis of pancreatitis based on CT
Cambridge 2 (Mild pancreatitis)	Two or more of the following: Main pancreatic duct (MPD) between 2–4 mm measured in the body, mild enlargement of the gland, heterogenic structure of the parenchyma, small cystic lesions (<10 mm), irregular ductal contour, more than three pathologically dilated side branches
Cambridge 3 (Moderate pancreatitis)	Same as 2 + MPD > 4 mm
Cambridge 4 (Severe pancreatitis)	Same as 2 and 3 + one of the following: cystic lesion >10 mm, parenchymal calcifications, ductal calcifications, ductal strictures, deformations of MPD

**Table 3 diagnostics-12-00841-t003:** SWV values in the CP group.

Results	SWV Head m/s	SWV Body m/s	SWV Tail m/s
$⚬\;\mathrm{Mean}\;\bar{X}$	1.85	1.76	1.63
⚬ Standard deviation	0.49	0.46	0.34
**Reference values** **(Robust method CLSI C28-A3^б^)**			
⚬ Lower limit (m/s)	1.03	1.11	0.88
⚬ Upper limit (m/s)	2.98	2.99	2.31

**Table 4 diagnostics-12-00841-t004:** SWV values in the CP group (entire parenchyma).

Descriptive Statistics	SWV Entire Parenchyma (m/s)
⚬ Lowest value	1.13
⚬ Highest value	2.81
$⚬\;\mathrm{Mean}\;\mathrm{value}\bar{X}$	1.75
⚬ Median	1.73
⚬ Standard deviation	0.34
**Reference values** **(Robust method CLSI C28-A3^б^)**	
⚬ Lower limit (m/s)	0.99
⚬ Upper limit (m/s)	2.43

**Table 5 diagnostics-12-00841-t005:** SWV values in the PDA group.

SWV Tumor (m/s)	Total	Male	Female	*p*-Value
Mean ± SD				0.009 **
	2.92 ± 0.91	2.55 ± 0.72	3.30 ± 0.95	
Min.–Max.				
	0.88–4.65	1.14–4.20	0.88–4.65	
Lower limit (m/s)	1.01	0.96	1.36	
Upper limit (m/s)	4.85	4.10	5.51	

Mean ± SD value for the tumor SWV irrespective of sex was established at 2.92 ± 0.9 m/s with reference values of 1.01 m/s to 4.85 m/s. **—significant association (*p* < 0.01).

**Table 6 diagnostics-12-00841-t006:** Accuracy, Se, and Sp of pSWE for differentiating between CP and PDA.

PDA→CP	AUC (95% CI)	*p*-Value	Sensitivity (95% ДИ)	Specificity (95% CI)	Cut-Off Value (m/s)
**SWV**	**0.886**		**89.47%**	**91.43%**	
	0.796 to 0.976	<0.001	75.20 to 91.10	76.90 to 98.20	**>2.09**

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
