# Peer review of "Evaluation of Ultrasound-Based Point Shear Wave Elastography for Differential Diagnosis of Pancreatic Diseases"

_diagnostics, 2022, doi:10.3390/diagnostics12040841_

Round 1

Reviewer 1 Report

This study represents an interesting idea pf comparing pSWE for chronic pancreatitis and pancreatic cancer , proved histologically . The results are interesting , but there are several points to be reviewed:

  1. Exclusion criteria: recent episode of acute pancreatitis should be  mentioned
  2. Chronic pancreatitis diagnosis:  the  exclusive inclusion of severe chronic pancreatitis or moderate chronic pancreatitis without  pancreatic would be preferable 
  3. The exclusion of patients with cystic component of the disease is important
  4. The location of pancreatic tumor is not specified, especially when compared to CP patients
  5. Results section

-table 5 and 6 are very charged and difficult to be followed. 

-Table 3,4, 7 and 8 are  unnecessary, the information  should be included in the text

Author Response

I want to express my gratitude for reviewing our article. I consider your comments and inquiries well found and justified. I tried to respond to all of them as thoroughly as possible.

Response to Reviewer 1 Comments

Point 1: Exclusion criteria: recent episode of acute pancreatitis should be mentioned

Response 1: We mention in the inclusion criteria for CP group that the patient must not have an episode of pancreatitis at least three months prior neither have biochemical signs of exacerbation at the time of evaluation. At least three previous symptomatic episodes of pancreatitis were also included as inclusion criteria. Certainly our intention was to exclude patients with exacerbation of CP or with either isolated or recurrent episodes of acute pancreatitis. In the revised version I will surely clarify the inclusion and exclusion criteria.

Point 2: Chronic pancreatitis diagnosis: the exclusive inclusion of severe chronic pancreatitis or moderate chronic pancreatitis without pancreatic would be preferable.

Response 2: As a secondary endpoint in our study we evaluated the influence of the stage of CP (according to Cambridge classification) on the SWV values. We found that severe CP is strongly associated with higher SWV compared to mild and moderate pancreatitis (p<0.014). In order to perform such investigation we had to include patients with all stages of CP (Cambridge II to IV). Of note and I’ll include this in the revised version, patients with inadequate presentation of all parts of the pancreatic gland were excluded from analysis. Thus patients with sizable chronic fluid collections (pseudocysts, walled-off necrosis) were naturally eliminated from the consequent statistical evaluation. In particular, we established that adequate visualization of the head, body and tail of the pancreas was possible in 93.5%, 97.2% and 92.8% of the patients, which lead to the exclusion of 5 patients in total of whom 4 were with CP. Since the aim of the present article was to evaluate the diagnostic performance of pSWE for differentiation between CP and PDA we didn’t include most of this data in order not to overcharge the narrative unnecessarily.

Point 3: The exclusion of patients with cystic component of the disease is important

Response 3: My response is in the context of your previous inquiry. Since according to Cambridge classification patients with cystic lesions larger than 10mm are regarded as having severe (stage IV) CP, we couldn’t afford to exclude all cystic lesion (even small ones) per se. As I mentioned since the presence of large fluid collections was almost universally associated with inadequate presentation of the entire gland, such patients were eventually excluded from analysis based on this criterion.

Point 4: The location of pancreatic tumor is not specified, especially when compared to CP patients

Response 4: I consider your remark fully justified. I’ll include this data in the Results section of the revised version. A comparison with the existing knowledge for the tumor distribution will be performed in the Discussion section.

Point 5: Results section - table 5 and 6 are very charged and difficult to be followed. Table 3,4, 7 and 8 are  unnecessary, the information  should be included in the text.

Response 5: I consider those comments being of more technical nature. Considering your recommendations, corrections will be applied accordingly.

Reviewer 2 Report

I would like to congratulate the authors for this well written study regarding a frequent clinical problem to differentiate between PDA and CP. Addition of pSWE clearly enhances the diagnostic value over conventional US B-mode examination. I have 1 question.

Were there any patients excluded from PDA or CP groups due to inadequate visualization of the pancreas by means of conventional B-mode? If yes, how many?

Author Response

I want to express my gratitude for reviewing our article. I think your inquiry was well founded and I tried to answer accordingly. Please see the attachment

Reviewer 3 Report

This paper descibes the effectiveness of  Shear Wave Elastography by abdominal ultrasonography. The study design is acceptable and the results are scientifically evaluated. My only concern is that ultrasonography has a limitation for observing the pancreas in fat peaple. 

Author Response

I wish to express my gratitude for reviewing our article. I consider your remarks well founded and I'll address them accordingly in the revised version. Please see the attachment.

Round 2

Reviewer 1 Report

I agree with the current form of changes.